# Supporting adolescents' participation in muscle-strengthening physical activity: protocol for the 'Resistance Training for Teens' (RT4T) hybrid type III implementation–effectiveness trial

Hayden Thomas Kelly [1,2,3] Jordan James Smith,[1,2,3] Angeliek Verdonschot [1,2,3] Sarah Grace Kennedy [4] Joseph J Scott,[5,6] Heather McKay,[7,8] Nicole Nathan [3,9,10,11] Rachel Sutherland,[3,9,10,11] Philip James Morgan [1,2,3] Jo Salmon [12] Dawn Penney [6] James Boyer [13] Rhodri S Lloyd [14] Christopher Oldmeadow,[3,10] Penny Reeves,[3,10] Kirrilly Pursey,[15,16] Myna Hua,[17] Sarah Longmore,[18] Jennifer Norman [19,20] Alexander Voukelatos [21] Avigdor Zask [22,23] David Revalds Lubans [1,2,3,24]

For numbered affiliations see end of article.

**Correspondence to**
Professor David Revalds Lubans;
David.Lubans@newcastle.edu.au

## ABSTRACT

**Introduction** In Australia, only 22% of male and 8% of female adolescents meet the muscle-strengthening physical activity guidelines, and few school-based interventions support participation in resistance training (RT). After promising findings from our effectiveness trial, we conducted a state-wide dissemination of the '*Resistance Training for Teens*' (RT4T) intervention from 2015 to 2020. Despite high estimated reach, we found considerable variability in programme delivery and teachers reported numerous barriers to implementation. Supporting schools when they first adopt evidence-based programmes may strengthen programme fidelity, sustainability, and by extension, programme impact. However, the most effective implementation support model for RT4T is unclear.

**Objective** To compare the effects of three implementation support models on the reach (primary outcome), dose delivered, fidelity, sustainability, impact and cost of RT4T.

**Methods and analysis** We will conduct a hybrid type III implementation–effectiveness trial involving grade 9 and 10 (aged 14–16 years) students from 90 secondary schools in New South Wales (NSW), Australia. Schools will be recruited across one cohort in 2023, stratified by school type, socioeconomic status and location, and randomised in a 1:1:1 ratio to receive one of the following levels of implementation support: (1) 'low' (training and resources), (2) 'moderate' (training and resources+external support) or 'high' (training and resources+external support+equipment). Training includes a teacher workshop related to RT4T programme content (theory and practical sessions) and the related resources. Additional support will be provided by trained project officers from five local health districts. Equipment will consist of a pack of semiportable RT equipment (ie, weighted bars, dumbbells, resistance bands and inverted pull up bar stands) valued

## STRENGTHS AND LIMITATIONS OF THIS STUDY

⇒ Our study will test three implementation support models on a range of implementation outcomes and determinants using a hybrid type III implementation–effectiveness trial.

⇒ Local health district staff (ie, project officers) are well placed to support the implementation of school-based interventions, but few studies have examined their capacity to support secondary schools.

⇒ There is potential for secondary school teachers to deliver Resistance Training for Teens with poor fidelity.

⇒ Our study does not include a control condition, but the low support group will serve as a 'usual practice' comparator.

⇒ Student-level data will only be collected in a subset of schools by secondary teachers, which may compromise internal validity.

at ~$A1000 per school. Study outcomes will be assessed at baseline (T0), 6 months (T1) and 18 months (T2). A range of quantitative (teacher logs, observations and teacher surveys) and qualitative (semistructured interviews with teachers) methods will be used to assess primary (reach) and secondary outcomes (dose delivered, fidelity, sustainability, impact and cost of RT4T). Quantitative analyses will use logistic mixed models for dichotomous outcomes, and ordinal or linear mixed effects regression models for continuous outcomes, with alpha levels set at p<0.025 for the outcomes and cost comparisons of the moderate and high support arms against the low support arm.

**Ethics and dissemination** Ethics approval has been obtained from the University of Newcastle (H-2021-0418), the NSW Department of Education (SERAP:2022215),

Hunter New England Human Research Ethics Committee (2023/ETH00052) and the Catholic Schools Office. The design, conduct and reporting will adhere to the Consolidated Standards of Reporting Trials statement, the Standards for Reporting Implementation Studies statement and the Template for Intervention Description and Replication checklist. Findings will be published in open access peer-reviewed journals, key stakeholders will be provided with a detailed report. We will support ongoing dissemination of RT4T in Australian schools via professional learning for teachers.

**Trial registration number** ACTRN12622000861752.

## INTRODUCTION

Physical inactivity has been described as a global health issue and is the fourth leading cause of premature death worldwide.[1] The WHO recommends children and adolescents participate in 60 min of (predominantly aerobic) moderate-to-vigorous physical activity (MVPA) daily, as well as muscle strengthening activity at least 3 days per week.[2] Despite these recommendations, physical activity declines substantially throughout adolescence, resulting in many adolescents failing to meet daily physical activity guidelines.[3] Fewer than 1 in 50 Australian adolescents (15–17 years old) meet MVPA and muscle strengthening activity recommendations.[4] Poor rates of muscle strengthening activity may explain the secular decline in muscular fitness (ie, strength, power and endurance) of Australian youth over the past 30 years,[5] with a similar decline seen internationally.[6 7] Muscular fitness is associated with various markers of health and well-being among children and adolescents in both cross-sectional[8] and prospective[9] studies. This prompted our team to investigate the key role of muscle strengthening activity during the school years.

Resistance training (RT) is a specialised form of muscle strengthening activity. It can be performed with or without equipment using a variety of resistive loads, and in a range of settings (eg, at home, school, local park or gym/fitness centre).[10] When performed routinely, RT may lead to muscle hypertrophy (ie, increase in muscle size), improved muscular fitness (ie, muscle strength, power and/or endurance), body composition (ie, increases in fat-free mass and reductions in fat mass) and mental health (ie, self-esteem) in school-aged youth.[11] RT is also recommended within global physical activity guidelines for adults,[1] as it benefits a wide range of physical and mental health outcomes (including risk of chronic disease and depression/anxiety). However, the performance of RT is often perceived as more complex than aerobic activities (eg, walking and jogging). This may explain why participation in RT is much lower than for MVPA during adulthood.[2] Gaining the knowledge, skills and confidence to participate in RT sets children and adolescents up for participation in RT across the lifespan.[12] However, for decades, a number of myths and misconceptions about the safety and appropriateness of strength training have prevented it from being offered to most school-aged youth.[12] Furthermore, among adults, there are a number of reported barriers to participation in RT, including lack of confidence,[13] low self-efficacy,[13] lack of time[14] and perceived lack of access to necessary equipment.[14] These barriers may also be present for adolescents, as they likely lack the knowledge, skills and/or confidence to engage in RT despite their desire to try a broader range of physical activities.[15] It is therefore necessary that adolescents are provided with ample opportunities to develop the skills, knowledge and confidence to engage in RT.

Schools are effective settings for health promotion as they provide access to most children and adolescents, and have the facilities, equipment and qualified staff needed to deliver interventions.[16–18] However, few school-based interventions provided adolescents with the confidence and competence to participate safely in RT[19] and no prior studies have done so at scale.[19 20] 'Scale-up' is defined as the 'deliberate effort to increase the impact of successfully tested health interventions to benefit more people'.[21] Few physical activity interventions have progressed beyond efficacy testing and no studies investigating scaled up RT interventions among secondary students appear within the extant literature.[22] The WHO has advised policy-makers, funders and researchers to focus their efforts on scaling up evidence-based physical activity programmes while exploring various models of implementation.[23] Prior to investing in scale-up, evaluating the effectiveness of models of implementation to inform future scale-up is necessary to guide the efficient use of resources. 'Voltage drop' refers to diminishing effectiveness when programmes are implemented at larger scale.[24 25] Evaluating different models of implementation will help to minimise this phenomenon.

Resistance Training for Teens (RT4T) is an evidence-based programme designed to provide adolescents with foundational knowledge about RT and enhance competence, confidence and motivation to engage in RT across the lifespan. It was also designed to align with the Australian Curriculum for Health and Physical Education to develop the movement skills and concepts that will enable students to participate in physical activities that contribute to their health and well-being.[26] We conducted a rigorous evaluation of the RT4T intervention using a cluster randomised controlled trial in 16 schools (n=607) between 2015 and 2016.[27] The intervention resulted in immediate and sustained improvements in upper body muscular fitness and RT skill competency, demonstrating an effective and scalable approach to delivering RT within secondary schools. During this period, we established a partnership with the New South Wales (NSW) Department of Education to scale-up the programme. With support from the NSW Department of Education, 468 teachers from 249 NSW Government schools were trained to deliver the programme.[27] We estimate that the programme reached ~10 000 students between August 2015 and October 2020. However, interviews conducted with a sample of teachers (n=19) who delivered RT4T, identified considerable variability in programme delivery. Also, several barriers to implementation emerged that included lack of support,

motivation and time,[28] impacting on programme fidelity and potential sustainability.

Implementation support, typically used for school physical activity programmes, is often criticised as being a 'train and forget' model. It comprises initial training workshops followed by limited ongoing support for schools. This approach is unlikely to address the primary impediments to programme implementation, but may explain some shortcomings in whether evidence-based health promotion programmes are adopted, implemented or maintained within schools. The most effective approach that optimises implementation fidelity and may sustain efficacious programmes over the longer term remains unclear.[29] Therefore, the aim of the present trial is to evaluate the effects of three implementation support models on the reach, effectiveness, dose delivered, fidelity, adoption, sustainability, impact and cost of RT4T.

## METHODS

### Experimental design

We will use a hybrid type III implementation–effectiveness trial design[30] to evaluate the implementation and effectiveness of the RT4T intervention. We aim to recruit 90 secondary schools and randomise them to one of three groups: (1) low, (2) moderate or (3) high implementation support. The low support group will act as a control group (usual practice). This is the professional development model traditionally used by the NSW Department of Education and the delivery model used for RT4T from 2015 to 2020. In Australia, the academic year is separated into four 'terms' of 10 weeks duration. Following training, teachers enrolled in our study will deliver the 8-week RT4T intervention within one (or more) of two school terms (ie, within 6 months post-training). This provides teachers with some flexibility regarding the delivery period for RT4T within their academic year.

Regarding the timeline for the evaluation of the intervention, study assessments are undertaken at baseline, 6-month and 18-month follow-up. Teachers will be encouraged to continue delivering the programme after our research has concluded. Our implementation trial is guided by expert recommendations[31] and is registered with the Australian New Zealand Clinical Trials Registry (ACTRN12622000861752). The design, conduct and reporting will adhere to the Consolidated Standards of Reporting Trials[32] statement, the Standards for Reporting Implementation Studies[33] statement and the Template for Intervention Description and Replication checklist.[34]

### School recruitment and selection

Secondary schools (government, Catholic and independent) in five local health districts in NSW (Illawarra Shoalhaven, Northern NSW, South Eastern Sydney, Sydney and Western NSW) will be eligible to participate in this study. Schools outside these districts will be able to attend the professional learning workshops and implement RT4T, but will not be included in the study to assess implementation outcomes. Eligible schools will be those that include students in grades 9 and 10 (aged 14–16 years). We will use a range of evidence-based recruitment and retention strategies to maximise participation and minimise dropout (at both school and teacher levels), including promotion within the NSW Department of Education School Sport Unit newsletter. This will include prenotification, the use of a dedicated recruitment coordinator, repeated reminders and deployment of staff from local health promotion teams to engage with schools.

### Participants

#### Teachers

One or two teachers from each enrolled school who teach students in grades 9 and 10 will be recruited to act as a 'school champion'. They will attend a single full-day professional learning workshop delivered by members of the research team with tertiary qualifications in physical education (PE), health promotion, exercise physiology and/or strength and conditioning. School champions will receive curriculum materials and resources at the workshop and will be asked to deliver a separate 2-hour training for other grade 9 and 10 teachers at their schools, who also have the option to implement the programme with their classes. Teachers can deliver the RT4T sessions to their students during usual PE lessons, co-curricular school sport periods or within an elective subject known as Physical Activity and Sports Studies. Professional learning will align with the NSW Educational Standards Authority accreditation process. Lastly, the learning will contribute to teachers' required annual training hours.[35 36]

#### Students

Secondary students in grades 9 and 10 (aged 14–16 years) enrolled in the study schools will be eligible to participate. The target population has been selected as this is the period during which students start to drop out of organised sport and can benefit from exposure to lifelong physical activities[37] such as RT.

### Primary and secondary outcomes

A range of quantitative and qualitative methods will be used to assess primary and secondary outcomes at baseline (T0), 6-month (T1) and 18-month follow-up (T2). Our primary outcome is reach, operationalised as the proportion of grade 9 and 10 students from the study schools who participate in the RT4T programme. The proportion will be calculated as a percentage of students from grades 9 and 10 who participate in ≥50% of the RT4T practical sessions divided by the total number of students in grades 9 and 10 at the study schools. Secondary outcome measures include dose delivered, fidelity, sustainability, impact, cost-effectiveness and implementation determinants.

### Sample size calculation

We aim to recruit 90 secondary schools, with 30 schools assigned to each of the three treatment arms. This will include an estimated 261 classes and n=~7800 students across the three arms of the trial. Based on our previous

research in secondary schools, we estimate 20% of schools will not provide usable data for our primary outcome at follow-up. As few studies have tested the effects of implementation strategies on the reach of the school-based physical activity interventions, there is little information to guide power calculations. Therefore, we estimated that the low support group will achieve 12.5% reach, while both the moderate and high support groups will achieve 25% reach (ie, between-group difference of 12.5%). Our power calculation is based on 90% power, type 1 error rate of 0.025 and SD of 12%. For the continuous secondary student effectiveness outcomes, we assumed an intraclass correlation coefficient estimate of 0.2. If 10% of schools provide usable data (~780 students), our study will have 80% power to detect significant small-to-moderate effects between the treatment arms.

### Blinding and randomisation

After teachers have completed the RT4T workshop, schools will be randomised to one of three implementation support arms. Schools will be stratified by school type, socioeconomic status and location and randomised using a random number producing algorithm by an independent statistical analysis service. Data analysis will be conducted by individuals blinded to group allocation.

### Evidence-based intervention

RT4T is an 8-week multicomponent physical activity programme, including practical and theory-based lessons designed to develop adolescents' knowledge, competence, confidence and motivation to participate in muscle strengthening activity. The design and delivery of RT4T is guided by the Supportive, Active, Autonomous, Fair and Enjoyable (SAAFE) principles for organised physical activity sessions for school-aged youth.[38] The training sessions and progressions were revised based on the Youth Physical Development Centre Basic Resistance Training Curriculum developed by Radnor *et al*.[39]

The RT4T programme includes four major components: (1) school-based RT sessions (delivered by teachers); (2) theoretical classroom sessions (delivered by teachers); (3) a smartphone and tablet app (used by students/teachers) and (4) energiser breaks (used by teacher during theoretical sessions).

### School-based RT sessions

Teachers will deliver the RT4T programme during their usual PE lessons, co-curricular school sport or Physical Activity and Sports Studies lessons. The RT4T practical sessions include four subcomponents: (1) movement based dynamic warm-up (5 min), (2) RT skill development (GymFit: 15–20 min) focused on five RT movement categories (ie, upper body push, upper body pull, lower body bilateral, lower body spilt and core stability), (3) choice of three different types of muscle strengthening activities: a fitness workout of the day (WOD), modified game with fitness infusion (GameFit), or a fun fitness challenge done to music (FunFit) (15–20 min); and

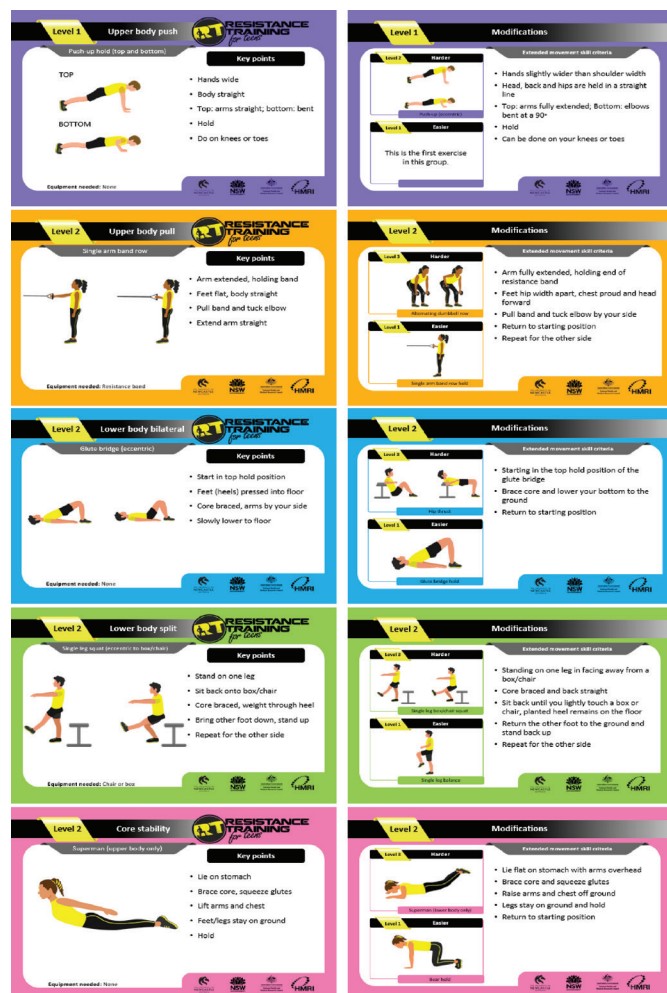

**Figure 1** Resistance Training for Teens exercise cards (front and back side).

(4) cool down and static stretching (StretchFit: 5 min). Practical sessions will be supported using a variety of resources, including exercise cards (see figure 1) and the RT4T smartphone and tablet app (see figure 2). Teachers will be also provided with a handbook that includes all the practical activities and theoretical content.

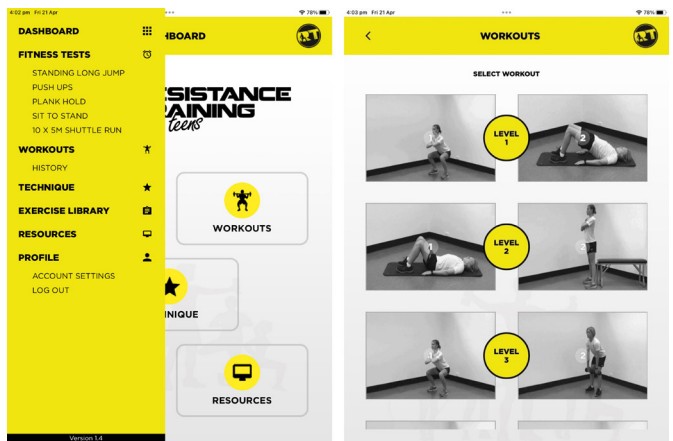

**Figure 2** Resistance Training for Teens app.

## Theoretical classroom-based lessons

Students will receive 8×40 min theory-based lessons that accompany weekly practical lessons. Content is purposefully aligned with the Australian curriculum for Health and Physical Education[40] and Australian Physical Literacy Framework.[41] The lesson topics are generally relevant to RT and adolescent health. Topics include: (1) introduction to RT, (2) fitness, self-assessment and self-monitoring, (3) fitness and e-Health technologies, (4) principles of training and RT programme design, (5) nutrition and RT: myths and recommendations, (6) social media, physical activity and body image, (7) yoga, Pilates and other community-based muscle strengthening activity and (8) reviewing fitness outcomes and realigning future goals.

## Smartphone and tablet app

We have developed and updated IOS and Android versions of the RT4T smartphone and tablet app to support teachers to implement the programme (see figure 2). Use of the app is optional and includes: (1) fitness testing, (2) evaluating RT skill competence, (3) pre-designed or customisable RT workouts and (4) an exercise library enabling users to browse bodyweight exercises organised by the five RT movement categories.

## Implementation strategies that support teachers to deliver RT4T

We will assess whether implementation strategies can be implemented effectively across three study arms that offer implementation support. We define implementation strategies as methods or techniques used to enhance adoption, implementation or sustainability of the evidence-based intervention.[42] Study arms are (1) low support or 'usual practice' (training and resources), (2) moderate support (training and resources+external support) and (3) high support (training and resources+external support+equipment) (see table 1). Implementation strategies were codeveloped with our partners in the NSW Department of Education and NSW local health districts and guided by findings from our previous dissemination study.[28] Our strategies align with the taxonomy of strategies as described by Proctor *et al*.[42]

## Implementation framework and strategies

Our approach is guided by the Consolidated Framework for Implementation Research (CFIR), which describes constructs across five domains: (1) intervention characteristics (RT4T), (2) outer settings (NSW Department of Education), (3) inner setting (schools), (4) individual (student/teacher) characteristics and (5) the process of implementation.[43] We describe relevant implementation strategies in table 1. We categorised implementation strategies as per the CFIR constructs and specified their usage using the Action, Actor, Context, Target, Time framework.[44]

### Professional learning for teachers

The school champion will complete a 30 min online course prior to programme delivery. This will include modules covering foundational principles of RT and specific safety recommendations to deliver RT to school-aged youth. Following initial completion of the online learning, the school champion will attend a single full-day workshop covering the background/rationale, structure and implementation of the RT4T programme, delivered by members of the research team with relevant expertise in PE, health promotion, exercise physiology and strength and conditioning.

School champions will be asked to attend the professional learning workshop and deliver a compressed training module to other grade 9 and 10 teachers within their schools. The compressed module comprises a train-the-trainer approach that is supported by resources (ie, presentation slides) provided by the research team. Our goal is to enable all interested and eligible teachers to deliver the RT4T programme to maximise the reach of the RT4T programme across the eligible student population.

### Project officers

The project officers from five local health districts will participate in a full-day training delivered by the research team. The training will be focused on the following: (1) RT4T programme content, (2) strategies to support teachers, (3) project officer roles and responsibilities and (4) how to assess fidelity of the RT4T programme. Following the teacher workshop, schools randomised to the moderate and high support groups will receive external support from project officers. Project officers with relevant qualifications in PE, health promotion, exercise physiology and/or strength and conditioning will observe teachers' implementation of the RT4T practical sessions, provide ongoing support and address the various barriers to programme implementation.

### Materials and equipment

All participating schools will receive approximately 100 hardcopy exercise cards to use during practical sessions (see figure 1). Schools allocated to the high implementation support group will also receive a basic RT equipment pack. Each equipment pack will include a variety of low cost, versatile exercise equipment (ie, 4x resistance bands set, 4x 3 kg dumbbells, 4x 5 kg weighted bar and 2x pull up bar stand).

### Measures and data collection

Quantitative assessments will be conducted at baseline (T0), 6 months (post intervention; T1) and 18 months (1 year post intervention; T2). Teachers will complete an online survey (T0, T1 and T2). Personal Development, Health, and Physical Education Heads of Department (20%, n=18) and School champions (20%, n=18) will be invited to participate in face to face or virtual interviews (T1 and T2).

### Implementation evaluation

Implementation outcomes are the effects of deliberate actions to deliver new policies, programmes and services.[45 46]

**Table 1** Domains, constructs and strategies used in our three implementation support models

| Domains | Constructs | Strategies | Level of support | | |
|---|---|---|---|---|---|
| | | | Low | Moderate | High |
| RT4T intervention | Evidence strength/ quality | Findings from the RT4T cluster randomised controlled trial were used in promotional and training materials. | ✔ | ✔ | ✔ |
| | Adaptability | Flexible intervention delivery model (physical education, school sport or elective) requiring minimal access to equipment. | ✔ | ✔ | ✔ |
| | Design quality and packaging | Programme resources developed by a professional graphic designer. | ✔ | ✔ | ✔ |
| Outer setting | Partnerships and investment | Investment from, and partnership with, key stakeholders from education (ie, NSW Department of Education) and health (ie, NSW Health local health districts). | ✔ | ✔ | ✔ |
| | External policy and incentives | Professional learning accreditation with national-based and state-based educational standards authorities. Alignment with state school physical activity policy. | ✔ | ✔ | ✔ |
| Inner setting | School culture | School champions will run a 2-hour face-to-face professional development workshop for other teachers in their school on RT4T. Prior to this, the teachers will have completed a 30 min prelearning session on resistance training safety. In the 2-hour workshop, the school champion will:<br>▲ Present the evidence and the rationale of RT4T.<br>▲ Teachers will participate in RT4T activities, discuss the content and gain consensus on how they will deliver RT4T.<br>▲ The school champion will share and explain the resources they received at the school champion workshop. | ✔ | ✔ | ✔ |
| | Leadership engagement | School champions and project officers will have approximately five contacts (three online and two face to face). In these meetings, they will:<br>▲ Discuss the action plan which was created by the school champion during the full-day workshop.<br>▲ Discuss the outline of the 2-hour training content.<br>▲ Evaluate two RT4T practical sessions, observed by the project officer and delivered by the school champion or another teacher who completed the 2-hour training delivered by the school champion. | ✔ | ✔ | ✔ |
| | Resources | Schools will receive exercise technique and workout resources. These will be provided to the school champion at the teacher workshop. All materials will also be available on a website where teacher can upload all the materials they need to deliver the programme. | ✔ | ✔ | ✔ |
| | Equipment | Schools will be provided with a basic RT equipment pack (~$A1000). This includes 4×5 kg weighted bars, 4×3 kg dumbbells, 2× portable pull up bar stands and 4× sets of resistance bands. These will be delivered to the schools in the high support group following randomisation after the workshop. | | | ✔ |

Continued

 Kelly HT, *et al. BMJ Open* 2023;**13**:e075488. doi:10.1136/bmjopen-2023-075488

**Table 1** Continued

| Domains | Constructs | Strategies | Level of support | | |
|---|---|---|---|---|---|
| | | | Low | Moderate | High |
| Individuals | Capability, opportunity, motivation (teacher) | A full-day professional development workshop will be provided to school champions. This workshop will be accredited with the NSW Educational Standards Authority and counts towards teachers professional learning. This will be delivered by the research team (UoN) who are experts in resistance training and school interventions. These will be held for groups of schools in the local area convenient to their location. Teacher relief for the school champions will be provided by the School Sport Unit (government schools) and University of Newcastle (non-government schools) to the school to cover the cost of their attendance. Prior to attending the workshop, school champions will complete the 30 min online prelearning on resistance training safety. The full-day workshop training will cover the following:<br>▲ Rationale for RT4T (ie, prevalence of physical inactivity in youth, benefits of RT and myths/facts regarding participation in RT).<br>▲ Adaptability of RT4T.<br>▲ Skills and demonstrations of the movements.<br>▲ RT4T lesson content and alignment with the Personal Development, Health, and Physical Education syllabus.<br>▲ Roles and responsibilities of school champions, teachers, and project officers.<br>▲ Introduction to the RT4T app. | ✔ | ✔ | ✔ |
| | Perceived barriers (student) | Addresses student motivation using SAAFE principles.<br>▲ Teachers will learn to deliver RT4T using the SAAFE principles. | ✔ | ✔ | ✔ |
| Implementation process | Planning for implementation | Teachers will complete an action plan template to support implementation of RT4T in their school. The action plan will include:<br>▲ When they are going to train their staff and how.<br>▲ When they are going to get resources to teachers and how.<br>▲ Roles and responsibilities of the school champion and teachers.<br>▲ School champions will complete the action plan during the full-day professional development workshop. | ✔ | ✔ | ✔ |
| | Project officers | School champions will be connected to a project officer who will provide ongoing support to the school champions after they have attended the workshop. The project officer will:<br>▲ Attend in person meetings with the school champion to gain commitment.<br>▲ Email/phone/visit the school champions.<br>▲ Support teachers via a closed RT4T Facebook group, which is created by the project manager from RT4T.<br>▲ Complete the checklist as part of the project officer handbook, which is developed by the research team.<br>▲ Stay in touch with the project manager from RT4T and provide all the needed details that are requested by the research team (eg, keep track of the support provided and join the weekly 10-min drop in meetings). | ✔ | ✔ | ✔ |
| | Evaluation and feedback | The project officers will observe approximately two RT4T session per school and provide feedback on the RT4T structure via email.<br>▲ Observations will be conducted using a standardised checklist developed by the research team that assesses aspects of the RT4T session (eg, did the teachers conduct the warm-up?).<br>▲ Project officers will only provide feedback on the RT4T structure via email. | | ✔ | ✔ |

NSW, New South Wales; RT4T, Resistance Training for Teens; SAAFE, Supportive, Active, Autonomous, Fair and Enjoyable; UoN, University of Newcastle.

## Primary implementation outcome
### Reach
Our primary outcome is reach, operationalised as the proportion of grade 9 and 10 students from the study schools who participate in the RT4T programme. The proportion will be calculated as a percentage of students from grades 9 and 10 who participate in ≥50% of the RT4T practical sessions divided by the total number of students in grades 9 and 10. We will assess reach by collecting teacher logbooks (ie, attendance lists) at T1 and T2. We have used logbooks previously in school-based health promotion trials[47 48] and achieved high completion rates, high validity and reliability.[49] A research team member will request that school champions provide a copy of teacher logbooks.

## Secondary implementation outcomes
### Representativeness
We will collect representativeness data (ie, School Index of Community Socio-Educational Advantage value, indigenous students (%) and students with language backgrounds other than English (%)) at the school level using the My School website. This information will be reported descriptively.

### Dose delivered
Dose delivered is the number of RT4T practical and theory lessons delivered to students. Teachers will be asked to record this information in the RT4T teacher handbooks.

### Fidelity
Project officers will observe approximately two RT4T practical sessions delivered by a teacher per school (moderate and high support arms) (total observations n= ~120). The research team will observe approximately one RT4T practical session from 50% of the schools allocated to the low support arm (total observations n= ~15). Session fidelity will be assessed by completing an observation checklist that describes the RT4T structure and process (eg, did the teacher include a warm-up?) and to what extent they implemented the SAAFE principles (eg, teacher was supportive and promoted positive student interactions, with answer options ranging from 1='*strongly disagree*' to 4='*strongly agree*').

### Sustainability
We define sustainability as the extent to which the programme is embedded within school practices. It will be assessed in two ways. First, through an interview with the Personal Development, Health, and Physical Education, Head of Department (at T2). Second, school champions will be asked to complete the Provider Report of Sustainment Scale sustainability tool.[50] The tool is a brief, pragmatic and generalisable three-item measure for front-line service providers. It assesses evidenced-based practice in different settings and has been shown to be a valid measure of sustainability.

### Implementation determinants
We define implementation determinants as factors believed or empirically shown to influence implementation outcomes.[51] We will assess the following determinants: culture, acceptability, feasibility, adaptability, compatibility (appropriateness), dose (satisfaction), capability, opportunity and motivation using a teacher survey (~20 min). This survey has not been used previously; however, it is based on a body of literature per different implementation outcomes including (n=number of survey items): participation in muscle strengthening exercise (6)[52] culture (1),[53] acceptability (1), feasibility (1), compatibility (appropriateness) (1),[54] adaptability (3), dose (satisfaction) (2),[55] capability (4), opportunity (5) and motivation (4).[56] All school champions will complete the questionnaire and will advise other teachers delivering the programme within their school to do the same. The research team will send reminders for survey completion to all teachers at 6-month and 18-month follow-up.

## Impact evaluation
Student-level outcome data will be collected from a convenience sample of students who provide informed parental consent. We anticipate that 10% of the sample will provide usable data (n~780 students). Fitness testing, participation in muscle strengthening activity and RT skill competency will be assessed using the RT4T app. The tablet version of the app (adapted from the 'Burn 2 Learn' programme)[57] will be used by teachers to assess students' fitness, student's participation in muscle strengthening activity and RT skill competency. We will evaluate upper body muscular endurance using the 90° push-up test.[58] Isometric abdominal muscular endurance will be measured using the plank hold test.[59] Standing long jump will assess muscular power, 1 min sit to stand will assess lower body muscular endurance and the 10×5 m shuttle run will assess speed and agility. These fitness assessments are integrated into the RT4T programme and conducted by the teachers at the start (week 1) and at the end (week 8) of the 8-week intervention period.

## Patient and public involvement
The NSW Department of Health, NSW local health districts, secondary school teachers and students were involved in the design of the RT4T intervention.

## Statistical analysis
Statistical analyses will be conducted by an independent statistical analysis service—Clinical Research Design, Information Technology and Statistical Support from Hunter Medical Research Institute. Analyses of the primary (programme reach) and secondary outcomes will be conducted using logistic mixed models (SAS Institute, Cary, North Carolina, USA) for dichotomous outcomes, and ordinal or linear mixed effects regression models for continuous outcomes. The primary outcome (ie, reach) will be collected at the school level and assessed using t-tests. Alpha levels will be set at p<0.025

for the comparisons of the moderate and high support arms against the low support arm. If the p values for the differences between the moderate and high support arms with the low support arm reach this threshold, the moderate and high support arms will be compared at a 5% significance threshold. For student-level outcomes, statistical analyses will be adjusted for the clustering of effects at the class level, as students from each school are nested in classes. Although clustering at the school level is negligible after adjusting for clustering at the class level, we will test this assumption and adjust our analyses for school-level clustering if required. For teacher-level outcomes, clustering will be accounted for at the school level. Two potential moderators of effects will be explored using interaction terms (ie, socioeconomic status of school and school location). School socioeconomic status will be determined using the Index of Community Socio-Educational Advantage (ICSEA), which will be obtained from the MySchool website.[60] School ICSEA values are determined using student-level parent occupation and education data, school location (ie, remoteness), and percentage of indigenous student enrolment.

### Data monitoring

All entered data will be deidentified using participant codes and stored in a password-protected drive at the University of Newcastle. Data will be checked for implausible values, and 20% of the data will be entered two times to confirm accuracy. It is not expected that participants will be exposed to greater risk of adverse events than they would be when participating in other types of school-based physical activity. However, the teacher handbook includes a section for teachers to report any adverse events that may occur. Any adverse events will be documented and reported to the relevant ethics committee. Any amendments to the study protocols will be publicly available via the Australian and New Zealand Clinical Trials Registry (trial number: ACTRN12622000861752).

### Economic evaluation

The economic evaluation will assess the costs and consequences of the RT4T programme across the three trial arms and will also include a budget impact analysis. The results of the cost–consequence analysis are presented as a scorecard comprising the total incremental cost of delivering the intervention alongside the range of outcomes reflected in the primary and secondary trial outcomes (consequences). This approach allows decision-makers to interpret the costs and outcomes of an intervention in a way that is relevant to their decision-making context.

Costs comprise the resource use associated with the intervention and implementation, while research costs are excluded. The opportunity cost for teacher and staff time will be prospectively measured and valued using pro rata Department of Education salary levels. The implicit cost of the spaces used to conduct the intervention will also be reported, as well as equipment costs. The budget impact analysis will be conducted to estimate the cost of scaling up the intervention across NSW and Australia. Consideration will be given to any cost offsets which would result from the scaling up of the intervention.

### ETHICS AND DISSEMINATION

Ethics approval has been obtained from the University of Newcastle (H-2021-0418), the NSW Department of Education (SERAP:2022215), Hunter New England Human Research Ethics Committee (2023/ETH00052) and the Catholic Schools Office. Students attending schools where the programme is delivered within the five local health districts will require parental consent for their fitness data to be recorded within the RT4T app. Students attending schools outside the five indicated local health districts will need to provide opt-out consent if they do not want their fitness data collected within the RT4T app. Our findings will be published in open access peer-reviewed journals. We will provide the NSW Department of Education, NSW Ministry of Health and all participating schools with a detailed report of our study findings. We will support ongoing dissemination of RT4T in NSW and beyond via a series of professional learning workshops.

### DISCUSSION

Few school-based interventions provide adolescents with the confidence and competence to participate safely in RT and no prior studies have done so at scale.[19] RT4T is an evidence-based programme designed to provide adolescents with the competence, confidence, knowledge and motivation to engage in RT throughout their lifetime. Importantly, RT4T was developed with scale-up in mind using the Consolidated Framework for Implementation Research (CFIR) to guide implementation.

Our study fills an important gap regarding the scale-up of effective school-based physical activity interventions.[61] Despite the large number of school-based efficacy and effectiveness trials, there is still only a small number of dissemination and implementation trials that have been conducted. As such, little is known regarding the strategies to support the implementation of physical activity programmes in schools. While the health benefits are well-established, there is a lack of school-based interventions supporting adolescents' safe participation in structured RT. This research will have widespread population health benefits as we leverage relationships through partnerships with local health district project officers and schools to increase programme scale-up.

However, there are limitations that should be noted. First, our study does not include a control group. However, the low implementation support group will serve as the RT4T 'usual practice' comparator. Second, our expanded delivery method is reliant on a 'train-the-trainer' model, and increasing the number of schools involved will help to determine feasibility across varying school contexts. Considering this, there is a risk of compromised fidelity to the intervention. Finally, regarding the effectiveness

evaluation, there is a risk of measurement bias influencing internal validity with only a subset of schools providing student-level data, which is to be collected by secondary teachers.

**Author affiliations**

[1]School of Education, University of Newcastle, Callaghan, New South Wales, Australia

[2]Centre for Active Living and Learning, College of Human and Social Futures, University of Newcastle, Callaghan, New South Wales, Australia

[3]Hunter Medical Research Institute, Newcastle, New South Wales, Australia

[4]School of Health Sciences, Translational Health Research Institute, Western Sydney University, Kingswood, New South Wales, Australia

[5]School of Education and Tertiary Access, University of the Sunshine Coast, Sippy Downs, Queensland, Australia

[6]School of Education, Edith Cowan University, Joondalup, Western Australia, Australia

[7]Department of Family Practice, University of British Columbia, Vancouver, British Columbia, Canada

[8]Active Aging Research Team, University of British Columbia, Vancouver, British Columbia, Canada

[9]Hunter New England Population Health, Hunter New England Local Health District, Wallsend, New South Wales, Australia

[10]School of Medicine and Public Health, College of Health, Medicine and Wellbeing, University of Newcastle, Callaghan, New South Wales, Australia

[11]National Centre of Implementation Science, University of Newcastle, Newcastle, New South Wales, Australia

[12]Institute for Physical Activity and Nutrition, Deakin University, Burwood, Victoria, Australia

[13]School Sport Unit, NSW Department of Education, Sydney, New South Wales, Australia

[14]Youth Physical Development Centre, Cardiff Metropolitan University, Cardiff, UK

[15]School of Health Sciences, Faculty of Health Medicine and Wellbeing, University of Newcastle, Callaghan, New South Wales, Australia

[16]Food and Nutrition Research Program, Hunter Medical Research Institute, New Lambton Heights, New South Wales, Australia

[17]Health Promotion Service, Population Health, South Eastern Sydney Local Health District, Sydney, New South Wales, Australia

[18]Health Promotion Service, Western NSW Local Health District, Bathurst, New South Wales, Australia

[19]School of Health and Society, Faculty of the Arts, Social Sciences and Humanities, University of Wollongong, Wollongong, New South Wales, Australia

[20]Health Promotion Service, Illawarra Shoalhaven Local Health District, Wollongong, New South Wales, Australia

[21]Population Health Research and Evaluation Hub, Sydney Local Health District, Forest Lodge, New South Wales, Australia

[22]Health Promotion, Northern NSW Local Health District, Lismore, New South Wales, Australia

[23]North Coast University Centre for Rural Health, School of Public Health, University of Sydney, Sydney, New South Wales, Australia

[24]Faculty of Sport and Health Sciences, University of Jyväskylä, Jyväskylä, Finland

**Contributors** HTK: software, resources, writing—initial draft and writing—review and editing. JJSm: conceptualisation, methodology, software, resources, writing—initial draft, writing—review and editing, supervision and funding acquisition. AVe: methodology, software, resources, writing—initial draft, writing—review and editing, supervision and project administration. SGK and JJSc: conceptualisation, methodology, resources, writing—original draft, writing—review and editing and funding acquisition. HM: conceptualisation, methodology, software, writing—original draft, writing—review and editing and funding acquisition. NN, RS, PJM, JS and DP: conceptualisation, methodology, writing—original draft, writing—review and editing and funding acquisition. JB: conceptualisation, methodology, writing—review and editing and funding acquisition. RSL and KP: methodology, resources and writing—review and editing. CO: methodology and writing—review and editing. PR: methodology and review and editing. MH, SL, JN, AVo and AZ: writing—review and editing and project administration. DRL: conceptualisation,

methodology, software, resources, writing—original draft, writing—review and editing, supervision and funding acquisition.

**Funding** This project is funded by a National Health and Medical Research Council (NHMRC) Partnership Grant (APP2010866). DRL is funded by an NHMRC Senior Research Fellowship (APP1154507). NN is funded by the Medical Research Future Fund (MRFF) Investigator Grant (GS2000053). This project is supported and codesigned with the NSW Department of Education.

**Disclaimer** The study funders will have no role in data collection, analysis, interpretation, or writing nor will they have influence over the publication of findings.

**Competing interests** None declared.

**Patient and public involvement** Patients and/or the public were not involved in the design, or conduct, or reporting, or dissemination plans of this research.

**Patient consent for publication** Not applicable.

**Provenance and peer review** Not commissioned; externally peer reviewed.

**ORCID iDs**
Hayden Thomas Kelly http://orcid.org/0000-0002-4533-6945
Angeliek Verdonschot http://orcid.org/0000-0002-1519-3402
Sarah Grace Kennedy http://orcid.org/0000-0001-9804-616X
Nicole Nathan http://orcid.org/0000-0002-7726-1714
Philip James Morgan http://orcid.org/0000-0002-5632-8529
Jo Salmon http://orcid.org/0000-0002-4734-6354
Dawn Penney http://orcid.org/0000-0002-2000-8953
James Boyer http://orcid.org/0000-0002-0585-4238
Rhodri S Lloyd http://orcid.org/0000-0001-8560-1566
Jennifer Norman http://orcid.org/0000-0001-6344-2605
Alexander Voukelatos http://orcid.org/0000-0003-3131-761X
Avigdor Zask http://orcid.org/0000-0002-3888-7335
David Revalds Lubans http://orcid.org/0000-0002-0204-8257

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
