## [Reviewer comments · BMJ Open]

ARTICLE DETAILS

TITLE (PROVISIONAL)	Supporting adolescents' participation in muscle-strengthening physical activity: Protocol for the 'Resistance Training for Teens' (RT4T) hybrid type III implementation-effectiveness trial
AUTHORS	Kelly, Hayden; Smith, Jordan; Verdonschot, Angeliëk; Kennedy, Sarah; Scott, Joseph; McKay, Heather; Nathan, Nicole; Sutherland, Rachel; Morgan, Philip; Salmon, Jo; Penney, Dawn; Boyer, James; Lloyd, Rhodri S.; Oldmeadow, Christopher; Reeves, Penny; Pursey, Kirrilly; Hua, Myna; Longmore, Sarah; Norman, Jennifer; Voukelatos, Alexander; Zask, Avigdor; Lubans, David

VERSION 1 – REVIEW

REVIEWER	van Sluijs, Esther MRC Epidemiology Unit
REVIEW RETURNED	08-Jun-2023

GENERAL COMMENTS	This well-written manuscript describes the protocol for a high quality trial on the implementation of a school-based resistance training programme for adolescents. Overall, it gives a good justification for the need for the trial and provides a good level of detail on the methods. I have some suggestions for some further detail to ensure greater clarity. P9 – please explain the purpose of a hybrid type III implementation-effectiveness trial and why this design was most optimal to address the research question. P11 – sample size. I was confused by the sample size calculation. On p20 the authors describe that 'reach' (the primary outcome) is assessed at school-level and it is unclear to me what the ICC of 0.2 is referring to for this outcome. This should be clarified. In addition, the authors should provide further clarification throughout as to the level at which the figures refer to (i.e. do you expect 30% school-level attrition). What is the 12.5% reach for the Low arm based on? Is 25% reach for the High arm considered 'effective'? Please justify. While I appreciate that sample size calculations are inherently imprecise, further justification should be provided as to the 'best guess estimate' for student outcomes. I could not see an indication as to the target recruitment for students, this should be included. And shouldn't this include an ICC? P11-20: Intervention. It is good to see the intervention described in good detail using the TIDieR checklist. However, I think the focus might need to be more on the implementation intervention (the focus of the investigation) and less on the intervention in schools as the current set-up is somewhat confusing. The authors could consider giving a more brief overview of the student-facing interventions, possibly moving the information now provided on
--

	p11-14 to a supplementary file, given more prominence to the implementation intervention. P17: I was unclear why 'school culture' was not included as delivered to the 'low' group as it was earlier described that this was an integral part of the RT4T intervention (p10). Please clarify. P20: The description of the primary outcome is somewhat ambiguous. It refers to the "proportion and representativeness of G9/10 students attending schools with school champions who deliver the RT4T programme (≥50% of practical sessions)." The proportion of what? Is the proportion of students attending 50% of the sessions (which seems to be suggested later on in the paragraph)? It is odd that representativeness is also included – is reach defined as the combination of both? There is no mention of how representativeness will be assessed. Please rewrite and clarify. P21: Project Officers are assessing fidelity – are these the same project officers also involved in intervention delivery (e.g. p17)? How will some independence be achieved? Will interrater reliability of the observation be assessed? P21: please provide more detail on the implementation questionnaire. Will this be completed by all teachers. Has this been used before or what is it based on? P21: Impact evaluation: "randomly selected sample of students" – how will they be randomly selected. What is the target N of students (and schools, if relevant). Is it maybe more a convenience sample? P22: please describe how SES status of school and school location will be assessed and defined. Minor typographical queries: P9, experimental design, line 5 "...give it is – the professional...": this sentence doesn't work. P21: please define PRESS
--	--

REVIEWER	Liu, Qiaolan Sichuan University
REVIEW RETURNED	29-Jul-2023

GENERAL COMMENTS	This protocol aims to compare the effects of three implementation support models on reach, dose delivered, fidelity, sustainability, impact and cost of RT4T. It is a valuable health promotion project. The following contents need to be improved.  1. What is the specific implementation time of this project? For how many years? 2. The age group covered by this protocol is 14-16 years (Grades 9 and 10). Why limit this age group? What age group is this program for? 3. Does the evaluation include physical or mental health indicators? For example body mass index and depression. What are the more specific cost-effectiveness measures? Evaluation indicators need to be described in more detail. 4. The statistical models and methods for evaluating the effects must be further elaborated.
--

VERSION 1 – AUTHOR RESPONSE

Reviewer: 1

Dr. Esther van Sluijs, MRC Epidemiology Unit

Comments to the Author:

This well-written manuscript describes the protocol for a high quality trial on the implementation of a school-based resistance training programme for adolescents. Overall, it gives a good justification for the need for the trial and provides a good level of detail on the methods. I have some suggestions for some further detail to ensure greater clarity.

1.1: P9 – please explain the purpose of a hybrid type III implementation-effectiveness trial and why this design was most optimal to address the research question.

We demonstrated the effectiveness of the RT4T intervention using a cluster randomised controlled trial involving 16 schools and 607 students (Kennedy et al., 2018). In our subsequent dissemination study, we identified considerable variability in program delivery and numerous barriers to implementation. Therefore, we designed an implementation trial to test the effects of different implementation support models on a range of implementation outcomes and determinants. As noted by Curran et al. (2012), the primary aim of a hybrid type III trial is to determine the utility of an implementation intervention/strategy.

Kennedy SG, et al. Implementing resistance training in secondary schools: A cluster randomised controlled trial. *Med Sci Sports Exerc.* 2018;50:62-72.

Kennedy SG, et al. Implementing Resistance Training in Secondary Schools: An exploration of teachers' perceptions. *TJACSM*, 2018;3:12:85–96.

Curran GM, et al. Effectiveness-implementation hybrid designs: combining elements of clinical effectiveness and implementation research to enhance public health impact. *Med. Care* 2012;50:3: 217.

1.2: P11 – sample size. I was confused by the sample size calculation. On p20 the authors describe that 'reach' (the primary outcome) is assessed at school-level and it is unclear to me what the ICC of 0.2 is referring to for this outcome. This should be clarified. In addition, the authors should provide further clarification throughout as to the level at which the figures refer to (i.e. do you expect 30% school-level attrition). What is the 12.5% reach for the Low arm based on? Is 25% reach for the High arm considered 'effective'? Please justify. While I appreciate that sample size calculations are inherently imprecise, further justification should be provided as to the 'best guess estimate' for student outcomes. I could not see an indication as to the target recruitment for students, this should be included. And shouldn't this include an ICC?

As noted by the Reviewer, our primary outcome, reach (i.e., proportion of Grade 9/10 students from each school who receive the RT4T program), is collected at the school level. Our statistician (A/Prof Oldmeadow) operationalised reach as a dichotomous student level outcome (i.e., each student in each school received the program or not). Therefore, to compare treatment effects, we need to account for the school-level ICC. In hindsight, we believe that it is simpler (and easier) to collect this information at the school level (i.e., number of Grade 9/10 students who receive the program divided by the total number of Grade 9/10 students in the study schools). We have revised our power calculation to assess the effects of the interventions on reach at the school level. We have also

adjusted our predicted retention rate. Based on the nature of the study, we expect 20% of schools will complete the professional training, but not provide adequate follow-up data. Of note, many of our target schools are in low-income communities that experience high rates of staff turnover. This is an issue we have encountered in previous large-scale implementation trials (Riley et al., 2021; Kennedy et al., 2021).

We have revised the power calculation statement below to improve clarity.

Line 289-301: “We aim to recruit 90 secondary schools, with 30 schools assigned to each of the three treatment arms. This will include an estimated 261 classes and N~7,800 students across the three arms of the trial. Based on our previous research in secondary schools, we estimate 20% of schools will not provide usable data for our primary outcome at follow-up. As few studies have tested the effects of implementation strategies on the reach of the school-based physical activity interventions, there is little information to guide power calculations. Therefore, we estimated that the Low support group will achieve 12.5% reach, while the Moderate and High support groups will both achieve 25% reach (i.e., between group difference of 12.5%). Our power calculation is based on 90% power, type 1 error rate of 0.025, and standard deviation of 12%. For the continuous secondary student effectiveness outcomes, we assumed an intraclass correlation coefficient (ICC) estimate of 0.2. If 10% of schools provide useable data (~780 students), our study will have 80% power to detect significant small-to-moderate effects between treatment arms.”

Riley N, et al. Dissemination of thinking while moving in maths: Implementation barriers and facilitators. *TJACSM* 2021;6:1:e000148.

Kennedy SG, et al. Evaluating the reach, effectiveness, adoption, implementation and maintenance of the Resistance Training for Teens program. *Int. J. Behav* 2021;18:122.

1.3: P11-20: Intervention. It is good to see the intervention described in good detail using the TIDieR checklist. However, I think the focus might need to be more on the implementation intervention (the focus of the investigation) and less on the intervention in schools as the current set-up is somewhat confusing. The authors could consider giving a more brief overview of the student-facing interventions, possibly moving the information now provided on p11-14 to a supplementary file, given more prominence to the implementation intervention.

We have slightly reduced this section in response to the Reviewer’s comment. However, we believe it is important to describe the student-facing material in our methods section rather than a supplementary file for two reasons. First, readers will expect to see this information in the main text. Second, the quality of our resources has improved since previous trials, and we would like to highlight these adaptations.

1.4: P17: I was unclear why ‘school culture’ was not included as delivered to the ‘low’ group as it was earlier described that this was an integral part of the RT4T intervention (p10). Please clarify.

We thank the Reviewer for identifying this error. We have amended Table 1 to include school culture for the Low support group (line 405).

1.5: P20: The description of the primary outcome is somewhat ambiguous. It refers to the “proportion and representativeness of G9/10 students attending schools with school champions who deliver the RT4T programme (≥50% of practical sessions).” The proportion of what? Is the proportion of students attending 50% of the sessions (which seems to be suggested later on in the paragraph)? It is odd that representativeness is also included – is reach defined as the combination of both? There is no mention of how representativeness will be assessed. Please rewrite and clarify.

We thank the Reviewer for their comment. To clarify, reach and representativeness serve as co-primary outcomes, we have revised our description of our primary outcome to improve clarity.

Line 451-463: "Reach: Our primary outcome is reach, operationalised as the proportion of Grade 9 and 10 students from the study schools who participate in the RT4T program. Proportion will be calculated as a percentage of students from Grade 9 and 10 who participate in $\geq 50\%$ of the RT4T practical sessions divided by the total number of students in Grade 9 and 10 at the study schools. We will assess reach by collecting teacher logbooks (i.e., attendance lists) at T1 and T2. We have used logbooks previously in school-based health promotion trials (Nathan et al., 2022; Nathan et al., 2020) and achieved high completion rates, good validity, and reliability (Cradock et al., 2014). A research team member will request that School Champions provide a copy of teacher logbooks that describes their delivery of the RT4T program. We will collect representativeness data [(i.e., School Index of Community Socio-Educational Advantage value, indigenous students (%), and students with language backgrounds other than English (%)] at the school level using the My School website. This information will be reported descriptively."

Nathan, N., et al., Multi-strategy intervention increases school implementation and maintenance of a mandatory physical activity policy: outcomes of a cluster randomised controlled trial. *British Journal of Sports Medicine*, 2022. 56(7): p. 385-393.

Nathan, N.K., et al., Implementation of a school physical activity policy improves student physical activity levels: outcomes of a cluster-randomized controlled trial. *Journal of Physical Activity and Health*, 2020. 17(10): p. 1009-1018.

Cradock, A.L., et al., Impact of the Boston active school day policy to promote physical activity among children. *American Journal of Health Promotion*, 2014. 28(3_suppl): p. S54-S64.

1.6: P21: Project Officers are assessing fidelity – are these the same project officers also involved in intervention delivery (e.g. p17)? How will some independence be achieved? Will interrater reliability of the observation be assessed?

We thank the reviewer for bringing this to our attention. In our previous version we described our Project Officers as "Project Officers from five local health districts, NSW Department of Education staff, and members of the research team", which is incorrect. NSW Department of Education staff and members of the research team will not be involved in supporting the teachers. We have amended the text in our manuscript and define Project Officers now as follows:

Project Officers (i.e., Project Officers from five local health districts)

We would like to clarify the difference between the Project Officers and the research team in roles and responsibilities. The Project Officers participated in a full-day training delivered by the research team. The training day focused on the following: (i) RT4T program content, (ii) strategies to support teachers, (iii) Project Officer roles and responsibilities, and (iv) how to assess fidelity of the RT4T program. The Project Officers are not involved in any of this and will only support the teachers with implementation (e.g., answering questions via email/phone calls and observing RT4T sessions). We have not planned to conduct interrater reliability for the following reasons. First, the fidelity checklist is simple to use, with raters required to identify the presence/absence of core session components. Second, the Project Officers were provided with training on how to assess fidelity. Third, we do not have capacity to conduct an interrater reliability study, but we feel confident the tool is reliable given its use in our previous studies (Kennedy et al., 2018; Cohen et al., 2015; Lubans et al., 2021).

We have added the following to the manuscript:

Line 422-425: "The Project Officers (i.e., Project Officers from five local health districts) will participate in a full-day training delivered by the research team. The training will be focused on the following: (i) RT4T program content, (ii) strategies to support teachers, (iii) Project Officer roles and responsibilities, and (iv) how to assess fidelity of the RT4T program."

Kennedy SG, et al. Implementing resistance training in secondary schools: A cluster randomised

controlled trial. *Med Sci Sports Exerc* 2018;50:62-72.

Cohen K, et al. Physical activity and skills intervention: SCORES cluster randomized controlled trial. *Med Sci Sports Exerc* 2015;47:765-774.

Lubans, DR, et al. Time-efficient intervention to improve older adolescents' cardiorespiratory fitness: Findings from the 'Burn 2 Learn' cluster randomised controlled trial. *Br. J. Sports Med* 2021;55:751-758.

1.7: P21: please provide more detail on the implementation questionnaire. Will this be completed by all teachers. Has this been used before or what is it based on?

All teachers that attend the professional learning workshop (i.e., School Champions) will complete the baseline questionnaire. School Champions will then deliver a 2-hour professional learning workshop for other teachers in their school who are interested in teaching the RT4T program. These teachers will also be asked to complete the baseline questionnaire. All teachers that complete the baseline questionnaire will be asked to complete the follow-up questionnaire. The questionnaire consists of different scales that have been designed for implementation studies. The questionnaire has not been used before, however is based upon a body of literature with different implementation outcomes (Weiner et al., 2017; Kennedy et al., 2020; Keyworth et al., 2020; Shakespear-Druery et al., 2022). The questionnaire also includes barriers to implementation (Carlson et al., 2017).

We have added the following to the manuscript:

Line 490-497: "This survey has not been used previously, however is based upon a body of literature per different implementation outcomes including (n = number of survey items): participation in muscle strengthening exercise (6) (Shakespear-Druery et al., 2022) culture (1) (Carlson et al., 2017), acceptability (1), feasibility (1), compatibility (appropriateness)(1) (Weiner et al., 2017), adaptability (3), dose (satisfaction) (Kennedy et al., 2020), capability, opportunity, and motivation (Keyworth et al., 2020). All School Champions will complete the questionnaire and will advise other teachers delivering the program within their school to do the same. The research team will send reminders for survey completion to all teachers at 6-month and 18-month follow-up"

Shakespear-Druery J, et al. Muscle-Strengthening Exercise Questionnaire (MSEQ): an assessment of concurrent validity and test-retest reliability. *BMJ Open Sport Exerc* 2022;8:1:e001225.

Carlson JA, Engelberg JK, Cain KL, et al. Contextual factors related to implementation of classroom physical activity breaks. *Transl Behav Med* 2017;7:581-92.

Weiner BJ, et al. Psychometric assessment of three newly developed implementation outcome measures. *Implement Sci* 2017;12:1:108.

Kennedy SG, et al. Process evaluation of a school-based high-intensity interval training program for older adolescents: the burn 2 learn cluster randomised controlled trial. *Children* 2020;7:12:299.

Keyworth C, et al. Acceptability, reliability, and validity of a brief measure of capabilities, opportunities, and motivations ("COM-B"). *Br. J. Health Psychol* 2020;25:3:474-501.

1.8: P21: Impact evaluation: "randomly selected sample of students" – how will they be randomly selected. What is the target N of students (and schools, if relevant). Is it maybe more a convenience sample?

The Reviewer is correct, our sample is best described as a convenience sample as students will not be randomly selected. Data will only be included from students who have returned parent consent as requested by our ethics committee. As noted in our power calculation section, if 10% of schools provide useable data (~780 students), our study will have 80% power to detect significant small-to-moderate effects between treatment arms.

The below text has been added to the manuscript:

Line 499-502: "Student-level outcome data will be collected from a convenience sample of students who provide informed parental consent. We anticipate 10% of the sample will provide usable data (N ~780 students). Fitness testing, participation in muscle strengthening activity and RT skill competency will be assessed using the RT4T app."

Please also see amended ethics and dissemination below:

Line 583-585: "Students attending schools where the program is delivered within the five local health districts will require parental consent for their fitness data to be recorded within the RT4T app."

1.9: P22: please describe how SES status of school and school location will be assessed and defined.

We will use the MySchool website to determine school level socio-economic status using the Index of Community Socio-educational Advantage (ICSEA, 2020).

The below text has been added to the manuscript:

Line 530-534: "School socio-economic status will be determined using the Index of Community Socio-Educational Advantage (ICSEA), which will be obtained from the MySchool website. School ICSEA values are determined using student-level parent occupation and education data, school location (i.e., remoteness), and percentage of indigenous student enrolment."

Australian Curriculum Assessment and Reporting Authority. Guide to understanding the Index of Community Socio-Educational Advantage (ICSEA), my school, editor, 2020.

Minor typographical queries:

1.10: P9, experimental design, line 5 "...give it is – the professional...": this sentence doesn't work.

This has been updated.

Line 230-232: "The Low support group will act as a control group (usual practice). This is the professional development model traditionally used by the NSW Department of Education, and the delivery model used for RT4T from 2015-2020."

P21: please define PRESS

PRESS is an acronym for Provider Report of Sustainment Scale. PRESS is a validated tool for measuring program sustainability (Moullin et al., 2021). Sustainability is one of the several secondary outcomes being assessed within this study

The below text has been added to the manuscript:

Line 483-485: "The tool is a brief, pragmatic and generalisable 3-item measure for frontline service providers. It assesses evidenced based practice in different settings and has been shown to be a valid measure of sustainment."

Moullin JC, et al. Provider REport of Sustainment Scale (PRESS): development and validation of a brief measure of inner context sustainment. *Implement Sci* 2021;16:86.

Reviewer: 2

Prof. Qiaolan Liu, Sichuan University

Comments to the Author:

This protocol aims to compare the effects of three implementation support models on reach, dose delivered, fidelity, sustainability, impact and cost of RT4T. It is a valuable health promotion project. The following contents need to be improved.

2.1: What is the specific implementation time of this project? For how many years?

We thank the reviewer for their comment.

In Australia, the academic year is separated into four 'terms' of 10 weeks duration. Following training, teachers enrolled in our study will deliver the 8-week RT4T intervention within one (or more) of two school terms (i.e., within six months post-training). This provides teachers with some flexibility regarding the delivery period for RT4T within their academic year.

Regarding the timeline for the evaluation of the intervention (i.e., the aim of our study), study assessments are undertaken at baseline, 6-months and 18-months follow-up. Teachers will be encouraged to maintain delivering the program after our research has concluded.

The experimental design has been amended with the following;

Line 232-239: "In Australia, the academic year is separated into four 'terms' of 10 weeks duration. Following training, teachers enrolled in our study will deliver the 8-week RT4T intervention within one (or more) of two school terms (i.e., within six months post-training). This provides teachers with some flexibility regarding the delivery period for RT4T within their academic year.

Regarding the timeline for the evaluation of the intervention (i.e., the aim of our study), study assessments are undertaken at baseline, 6-months and 18-months follow-up. Teachers will be encouraged to maintain delivering the program after our research has concluded."

2.2: The age group covered by this protocol is 14-16 years (Grades 9 and 10). Why limit this age group? What age group is this program for?

The RT4T program would likely be suitable for younger students, however age 14-16 is a period within which rates of organised sports participation decline significantly and is also when adolescents are more receptive to learning about alternatives to sport such as resistance training. By limiting the program to this age group, we aim to reduce the variability in effectiveness between schools that might otherwise present due to greater variation in the age range of participating students. This will allow us to form firmer conclusions regarding the effectiveness of RT4T for 14-16 year olds who we argue stand to benefit more from this specific intervention than younger students who are typically more active.

The following has been added to the manuscript:

Line 275-277: "The target population has been selected as this is the period during which students start to drop out of organised sport and can benefit from exposure to lifelong physical activities (Hulteen et al., 2018) such as resistance training."

Hulteen RM, et al. Development of foundational movement skills: A conceptual model for physical activity across the lifespan. *Sports Med.* 2018;48:1533-1540.

2.3: Does the evaluation include physical or mental health indicators? For example body mass index and depression.

We will examine the impact of RT4T on student muscular fitness in a sub-sample of participants. We have previously demonstrated the benefits of RT4T for adolescents' upper body muscular fitness, resistance training skill competency, resistance training self-efficacy, and body mass index (Kennedy et al., 2018).

Kennedy, S.G., et al., Implementing resistance training in secondary schools: a cluster randomized controlled trial. *Medicine & Science in Sports & Exercise*, 2018. 50(1): p. 62-72.

2.4: What are the more specific cost-effectiveness measures? Evaluation indicators need to be described in more detail.

The following text has been amended:

Line 546-559: "The economic evaluation will assess the costs and consequences of the RT4T

program across the three trial arms and will also include a budget impact analysis. The results of the cost-consequence analysis are presented as a scorecard comprising the total incremental cost of delivering the intervention alongside the range of outcomes reflected in the primary and secondary trial outcomes (consequences). This approach allows decision makers to interpret the costs and outcomes of an intervention in a way that is relevant to their decision-making context. Costs comprise the resource use associated with the intervention and implementation, while research costs are excluded. The opportunity cost for teacher and staff time will be prospectively measured and valued using pro-rata Department of Education salary levels. The implicit cost of the spaces used to conduct the intervention will also be reported, as well as equipment costs. The budget impact analysis will be conducted to estimate the cost of scaling up the intervention across NSW and Australia. Consideration will be given to any cost offsets which would result from the scaling up of the intervention.”

2.5: The statistical models and methods for evaluating the effects must be further elaborated.

We have included a detailed statistical analysis section below:

Line 514-534: “Statistical analyses will be conducted by an independent statistical analysis service - Clinical Research Design, Information Technology and Statistical Support from Hunter Medical Research Institute. Analyses of the primary (program reach) and secondary outcomes will be conducted using logistic mixed models (SAS Institute Inc, Cary, NC) for dichotomous outcomes, and ordinal or linear mixed effects regression models for continuous outcomes. The primary outcome (i.e., Reach) will be collected at the school level and assessed using t-tests. Alpha levels will be set at $p < 0.025$ for the comparisons of the Moderate and High support arms against the Low support arm. If the p-values for the differences between the Moderate and High support arms with the Low support arm reach this threshold, the Moderate and High support arms will be compared at a 5% significance threshold. For student level outcomes, statistical analyses will be adjusted for the clustering of effects at the class level, as students from each school are nested in classes. Although clustering at the school level is negligible after adjusting for clustering at the class level, we will test this assumption and adjust our analyses for school-level clustering if required. For teacher-level outcomes, clustering will be accounted for at the school-level. Three potential moderators of effects will be explored using interaction terms (i.e., type of Project Officer, socio-economic status of school, and school location). School socio-economic status will be determined using the Index of Community Socio-Educational Advantage (ICSEA), which will be obtained from the MySchool website. School ICSEA values are determined using student-level parent occupation and education data, school location (i.e., remoteness), and percentage of indigenous student enrolment.”

VERSION 2 – REVIEW

REVIEWER	van Sluijs, Esther MRC Epidemiology Unit
REVIEW RETURNED	29-Sep-2023

GENERAL COMMENTS	The authors have adequately addressed the reviewers' comments. However, I have one remaining point for clarification: 1.5 - In your response you state "To clarify, reach and representativeness serve as co-primary outcomes, we have revised our description of our primary outcome to improve clarity." However, throughout the remainder of the response and the text you refer to 'reach' as the primary outcome and 'representativeness' is not explicitly mentioned as either a primary or a secondary outcome. Please clarify
--

REVIEWER	Liu, Qiaolan
-----------------	--------------

	Sichuan University
REVIEW RETURNED	25-Sep-2023
GENERAL COMMENTS	This protocol is much improved. There are no other comments.

VERSION 2 – AUTHOR RESPONSE

Reviewer Reports:

Reviewer: 1

Dr. Esther van Sluijs, MRC Epidemiology Unit

Comments to the Author:

The authors have adequately addressed the reviewers' comments. However, I have one remaining point for clarification:

1.5 - In your response you state "To clarify, reach and representativeness serve as co-primary outcomes, we have revised our description of our primary outcome to improve clarity." However, throughout the remainder of the response and the text you refer to 'reach' as the primary outcome and 'representativeness' is not explicitly mentioned as either a primary or a secondary outcome. Please clarify

Response:

We thank the reviewer for addressing this and we understand it may cause confusion to the reader. We have now listed 'representativeness' under 'secondary outcomes' and have 'reach' as our only main outcome to clarify, as follows:

Line 384-400:

"Primary implementation outcome

We define an implementation outcome as the intentional actions to deliver a policy or an intervention [45, 46].

Reach: Our primary outcome is reach, operationalised as the proportion of Grade 9 and 10 students from the study schools who participate in the RT4T program. Proportion will be calculated as a percentage of students from Grade 9 and 10 who participate in $\geq 50\%$ of the RT4T practical sessions divided by the total number of students in Grade 9 and 10 a. We will assess reach by collecting teacher logbooks (i.e., attendance lists) at T1 and T2. We have used logbooks previously in school-based health promotion trials [47, 48] and achieved high completion rates, good validity, and reliability [49]. A research team member will request that School Champions provide a copy of teacher logbooks that describe details of the RT4T program.

Secondary implementation outcomes

Representativeness: We will collect representativeness data [(i.e., School Index of Community Socio-Educational Advantage value, indigenous students (%), and students with language backgrounds other than English (%)) at the school level using the My School website. This information will be reported descriptively."